# Patients’ Engagement in Early Detection of COVID-19 Symptoms: An Observational Study in the Very Early Peak of the Pandemic in Italy in 2020

**DOI:** 10.3390/ijerph19053058

**Published:** 2022-03-05

**Authors:** Lorenzo Palamenghi, Fabiola Giudici, Guendalina Graffigna, Daniele Generali

**Affiliations:** 1Engageminds HUB-Consumer, Food and Health Engagement Research Center, Università Cattolica del Sacro Cuore, 20123 Milano, Italy; guendalina.graffigna@unicatt.it; 2Department of Psychology, Università Cattolica del Sacro Cuore, 20123 Milano, Italy; 3Faculty of Agriculture, Food and Nutritional Sciences, Università Cattolica del Sacro Cuore, 26100 Cremona, Italy; 4Biostatistics Unit, Department of Medical Sciences, University of Trieste, 34127 Trieste, Italy; fgiudici@units.it; 5Breast Cancer Unit, ASST of Cremona, Viale Concordia 1, 26100 Cremona, Italy; daniele.generali@unicatt.it; 6Department of Medical Surgery and Health Sciences, University of Trieste, 34127 Trieste, Italy; 7Department of Animal Science, Food and Nutrition–DIANA, Università Cattolica del Sacro Cuore, 29122 Piacenza, Italy

**Keywords:** COVID-19, general practitioner, symptoms, public health, health engagement

## Abstract

COVID-19 exerted a strong impact on the Italian healthcare systems, which in turn resulted in a reduction in the citizens’ trust towards healthcare authorities. Moreover, the focused attention on the typical COVID-19 symptoms (fever, cough) has also impacted the social representation of health priorities, potentially reducing the perceived importance and severity of other symptoms. This study aimed to determine the association of general-practitioner (GP) contact with various symptoms during the COVID-19 pandemic in Cremona, an Italian city at the very epicentre of the pandemic. Between April and June 2020, an anonymous survey was completed by 2161 respondents. Logistic-regression analyses were used to examine the associations of GP contact with sociodemographic characteristics and the presence of symptoms. Of the 2161 respondents (43.5% female, 75.0% aged less than 55 years), 959 (44.4%) reported experiencing various symptoms and 33.3% contacted a GP. GP contact was significantly associated with poor appetite (OR, 2.42; 95% CI 1.63 to 3.62; *p* < 0.001), taste dysfunctions (OR 1.67; 95% CI 1.20 to 2.34; *p* < 0.001) and sleepiness during the day (OR 4.15; 95% CI 2.13 to 8.09; *p* = 0.002). None of the gastrointestinal symptoms resulted in significantly increasing the likelihood of contacting a GP. This study offers a unique observation of citizens’ attitudes and behaviours in early symptom communication/detection during the initial peak of the Italian COVID-19 pandemic.

## 1. Introduction

The COVID-19 pandemic exerted a strong impact on healthcare systems, both from a financial [1] and an organizational point of view [2]. In particular, regardless of their dedication and effort, healthcare operators all over the world found difficulty in coping with COVID-19-related distress due to the increased job demands [3]. Among the various consequences that these increased demands had on healthcare operators’ health and work performance [4], recent studies showed that due to these increased demands, healthcare workers had to reduce the attention they could pay to diseases other than COVID-19 [5,6].

Furthermore, from the perspective of the public opinion, the media narrative that has strictly focused on the COVID-19 emergency has also impacted the social representation of health priorities, which in turn has changed patients’ attitudes towards their disease management and their perception of health risks. The emotional impact of the pandemic, which is often correlated to the feeling of paralyzing fear, an increased sense of uncertainty, depressive reaction and fatalism [7,8], has furthermore contributed to a decrease in the level of patients’ activity in their self-management and in their beliefs in being able to control risk factors related to their health condition [9,10]. Indeed, this decreased sense of agency and self-efficacy in the management of one’s own health and lifestyle is a preoccupying element, as it may undermine the citizens’ level of vigilance of their own symptoms and health status, as well as their willingness to promptly communicate with their healthcare professionals. However, it is now widely agreed by the international scientific community that in order to guarantee effective healthcare it is important to establish a partnership with the patients for the early communication of symptoms [11,12,13].

The combination of the decreased attention of healthcare workers and the reduced activity of citizens has the disruptive potential to impact prevention and screening methods, potentially leading to an increase in non-communicable-disease prevalence or severity [14,15]. The risk is that the attention and effort put into the management of COVID-19 may drain resources from the management of other diseases [5,16], thereby hampering prevention, diagnosis of new-onset diseases, and medication adherence, potentially leading to a worsening of public health [17].

Indeed, the current pandemic situation has made the importance of citizens’ behavioural activity towards prevention and health self-management particularly relevant [18], as their own initiative for getting in contact with a healthcare provider in the case of various symptoms is crucial for an early intervention and for containing the spread of diseases other than COVID-19 [19,20]. This is particularly relevant for the Italian case, as Italy (and Lombardy, where this study was conducted, in particular) was the first epicentre of the COVID-19 pandemic in a western country, which resulted in a disruptive impact on a healthcare system that was structured in a very hospital-centred perspective, with little or no synergy with the citizens’ community [21]. Paradoxically, although the COVID-19 pandemic has cast light on the crucial importance of citizens’ engagement and their willingness to collaborate with the healthcare system in facing the emergency by adhering to the preventive measures, in practice the current management of the emergency has further decreased the alliance between citizens and their healthcare professionals [9,22]. Citizens have shown perplexities and difficulties in changing their behaviours and in trusting healthcare authorities, as well as in maintaining a good engagement trajectory for their self-management.

Thus, the aims of the present study were to:Assess the prevalence of people who decided to contact their general practitioner for a consultation in presence of various symptoms.Assess which symptoms are at a higher risk of being neglected.

Moreover, we were interested in comparing a group of laypeople (recruited among a group of persons with high attention to and regard towards healthcare, such as blood donors) and a group of healthcare professionals, in order to compare eventual differences in how they cope with symptoms.

## 2. Materials and Methods

### 2.1. Participants and Procedure

Between April and June 2020, an online survey was administered to a group of blood donors, a group of healthcare professionals, and a group of healthcare volunteers. Participants were recruited via a purposive sampling and contacted via email. All the participants were recruited in Cremona, Italy, which was one of the first cities, in Italy and in the western countries, to be struck by the COVID-19 pandemic. Before accessing the survey, the participants provided informed consent. 

The study was conducted in accordance with the Declaration of Helsinki, the Good Clinical Practice principles and all local regulations. The study obtained the approval of the ethical committee of the ASST of Cremona Hospital (IRB Approval 07/04/2021 n.11781/2021).

### 2.2. Materials

The online survey included a series of questions regarding: participants’ socio-demographic characteristics (i.e., gender, age, occupational status, with whom he/she lives), health-related information, smoking behaviour (past smoker, current smoker, never smoked), previous vaccinations (anti-flu in last autumn, pneumococcal in the last 12 months, other vaccinations in the past 12 months), presence of various symptoms (diarrhoea, nausea, lack of appetite, anosmia, loss of taste, vertigo, headache, sleepiness, confusion, tremors, face/arms/legs formication).

Participants who reported any symptom were then asked whether they were in touch with their general practitioner. 

### 2.3. Statistical Analyses

Descriptive statistics were used to characterize the overall study population (blood donors, professionals, volunteers) using absolute frequencies and percentages for categorical variables.

The primary outcome of this study was to identify any COVID-related symptoms, including diarrhoea, nausea/vomiting, poor appetite, smell dysfunctions, taste dysfunctions, dizziness, headache, sleepiness during the day, confusion, involuntary tremors, tingling in the limbs/face, that may prompt a subject to contact a general practitioner. The analysis regarding the primary outcome was only conducted in the subgroup of subjects who declared in the questionnaire to have had at least one of the previous symptoms (*n* = 959). The presence of every symptom was dichotomized as follows: “present vs. absent”, regardless of the number of days from which the symptom was observed. Logistic-regression models were used to estimate the independent effect of sociodemographic characteristics and of symptoms on the probability of contacting a GP (dependent variable).

The single-symptom analysis (i.e., diarrhoea yes/no) did not exclude that the subject may have other symptoms. For this reason, a variable “Number of Concomitant Symptoms” was also created in order to take this fact into account.

Variables with a *p*-value lower than 0.10 in the univariate analysis were considered for the multivariate models and adjusted for age and sex. Odd ratios (ORs) and 95% confidence intervals (95% CI) were reported in order to summarize the logistic-regression analysis, which was performed separately for the three groups of subjects (blood donors, professionals and volunteers). The results related to this last group should be considered only as a descriptive because the number of events (i.e., subjects who had contacted the practitioner) was too low for a statistical evaluation. All analyses were conducted using R statistical software (version 4.0.2 [23]). *p*-values < 0.05 were considered statistically significant.

## 3. Results

### 3.1. Sample Characteristics

Overall, 2193 potential participants were contacted. A total of 2161 (98.5%) gave informed consent and actually took part in the study. Of these, 1569 were blood donors (72.61%), 407 were professionals (18.83%), and 185 were volunteers (8.56%). A total of 940 participants (43.50%) were females. The ages of the participants ranged between 18 and 85 years old, with most participants in the 26–45-year-old group (38.45%). A total of 1718 participants (79.50%) reported living with someone, and 1802 (83.39%) reported that they were currently employed.

The majority of the respondents reported not smoking (65.57% reported never smoking, and an additional 18.97% reported having smoked in the past, but also having quit), and having a normal body-mass index (56.87%). Table 1 shows the participants’ characteristics in detail.

### 3.2. Reported Symptomatology

Overall, 959 interviewed participants (44.4%) reported at least one of the listed symptoms. In particular, the most frequent were headache (29.7%), sleepiness during the day (17.1%), and diarrhoea (14.2%). Looking into the subgroups, 659 blood donors (42.0%), 209 professionals (51.4%), and 91 volunteers (49.2%) reported having experienced at least one of these symptoms. Overall, 217 blood donors out of the 659 that experienced symptoms (32.9%) contacted a GP, 69 professionals out of 209 (33%), and 33 volunteers (47.82%).

### 3.3. Univariate and Multivariate Analyses

Table 2 and Appendix A show the association of the presence of symptoms and the influence of COVID-19 on the contact of a GP for the whole sample and for the three population subgroups (blood donors, professionals and volunteers), respectively.

The multivariate model applied to all cohort showed that poor appetite, taste dysfunction and sleepiness during the day were the symptoms that increased the likelihood of getting in touch with the GP. All other symptoms and sociodemographic factors resulted non-significant. Similar results were obtained considering the subgroups: the multivariate models applied to the professionals and blood donors’ groups confirmed that poor appetite and taste dysfunction increased the likelihood of getting in touch with the GP. Only for the blood donors we found that the smoking factor was associated to the outcome. Multivariate analyses for the volunteers group were not performed due to the limited sample size.

## 4. Discussion

Overall, only about one out of three participants reported contacting a GP after feeling sick. This data is quite surprising due to the peculiar epidemiological situation portrayed by the study: in early 2020, at the time of the data collection, Italy and in particular Cremona were at the epicentre of the pandemic, with a peak number of deaths and of hospitalized people and a general situation of psychological alarm shared by citizens and healthcare professionals alike [21,24,25,26]. Nevertheless, the general level of confusion and emotional distress may have led the local population to a less proactive collaboration with GPs. Given that the data from this survey were collected during the peak of the pandemic in Italy, it is also likely that this number might be due to the difficulties in mobility experienced by citizens while the lockdown measures were being implemented, as well as the fear of being infected by the virus by visiting clinics and hospitals [27], even though telephone consultations were generally available at the time. However, this may also be symptomatic of a lack of patient trust in their primary doctor and of the perception of hindrances in the direct communication with him/her [28,29]. Doctor-patient communication has been largely demonstrated to be a crucial factor of sustaining patient engagement in self-management [30,31,32]. Furthermore, a good perception of trust in one’s own primary doctor is considered the basis for sustaining patients’ healthier behaviours and their adherence to preventive prescriptions [33,34,35]. Particularly in an emergency situation, the collaboration between citizens and the healthcare system is needed in order to enable timely and effective actions, resulting in a robust trust relationship.

Generally speaking, the socio-demographic variables such as gender and age, as well as having a chronic condition, did not significantly increase the likelihood of contacting a GP. Indeed, the only exception was that, only among blood donors, former smokers and non-smokers were more likely to participate in the healthcare system compared to smokers. This agrees with the assumption that individuals who engage in healthier lifestyles (such as non-smokers) tend to have an enhanced psychological motivation for healthier behaviours [36] and prevention [37,38]. In other terms, people who engage less in risky health behaviours may be more inclined to actively take care of their wellness, and this may be the basis, even during an emergency such as the COVID-19 pandemic, for proactivity in symptom detection and better collaboration with healthcare providers. Similarly, the higher proactivity in symptom management of blood donators reported in this study may be linked to their enhanced attention to the common good: donors are often motivated by altruistic (rather than egoistic) values [39].

The symptoms that, overall, resulted in significantly increasing the likelihood of contacting a GP were poor appetite, taste dysfunction, dizziness, and sleepiness during the day (the latter two only for the blood-donor group, possibly due to the larger sample size). In addition to these symptoms, smell dysfunction, nausea/vomiting, confusion, involuntary tremors and diarrhoea were significant, although only in the univariate analyses. Interestingly, the fact that gastrointestinal symptoms did not show a particularly relevant impact on the likelihood of contacting a GP may indicate that, even though these symptoms are a possible manifestation of COVID-19 [40,41,42], there might be a low awareness in the population that this is indeed the case, when compared to symptoms such as scarce appetite, sleepiness and taste/smell dysfunctions.

Finally, a few symptoms did not result in overall significance, such as headache, which is another symptom that is potentially associated with COVID-19 [43,44]. Interestingly, in the univariate analysis, the number of overall symptoms resulted in increasing the likelihood of contacting a GP.

### Limitations

This study has a few limitations. First, the sampling was purposive and the sampled population (blood donors due to logistic constrains) might have some biases compared to the general population: for this reason, the generalization of the results described in this study should be made with some caution. Additionally, symptoms that are more typical of COVID-19 such as fever and cough were not assessed in our study.

## 5. Conclusions

This study offers a unique observation of citizens’, volunteers’, and healthcare professionals’ attitudes and behaviours in early symptom communication/detection during the initial peak of the Italian COVID-19 pandemic in the geographical area of Cremona (Italy). The data show the barriers and hindrances experienced by the three different groups in detection and timely communication of symptoms. These data are a potential indicator of a lack of engagement and proactivity in health prevention. Patient engagement has been largely agreed to be fundamental in order to guarantee patients’ effective self-management and adherence to preventive measures. The lack of patient engagement-particularly during a health crisis can cause poor collaboration between citizens and healthcare professionals, leading to disorganization, chaos, and the loss of timely intervention. However, high levels of patient engagement are a function of a trustful relationship between patients and their primary doctor. Furthermore, patient engagement needs to be fostered and sustained by an inclusive healthcare organization that integrates hospital services and community services and is able to put the patients and their needs at the centre. The critical evidence that emerged in this study thus raises warns of the potential inconsistency of patient engagement and collaboration between civil society and the healthcare system in the geographical area of Cremona, and suggests that this lack of collaboration may have been a further critical element that contributed to the dramatic spread of the pandemic in the area and to its tremendous impact in terms of lives, hospitalization and psychological trauma of the civil society. This evidence indicates the need for an increased consideration of the importance of engaging patients in their symptom detection and healthcare management. Moreover, it suggests the need for enhanced investment, not necessarily just during a healthcare crisis, in educational intervention aimed at fostering citizens’ awareness about their role in collaborating with the healthcare system to manage an emergency.

## Figures and Tables

**Table 1 ijerph-19-03058-t001:** Sample characteristics.

Variables	Cohort: *n* = 2161
**Group (N, %)**	
Blood donors	1569 (72.61%)
Healthcare Professionals	407 (18.83%)
Volunteers	185 (8.56%)
**Age (N, %)**	
18–25	153 (7.08%)
26–45	831 (38.45%)
46–55	637 (29.48%)
56–65	430 (19.90%)
66–75	102 (4.72%)
76–80	6 (0.28%)
81–85	2 (0.09%)
**Sex (N, %)**	
Female	940 (43.50%)
Male	1220 (56.5%)
No answer	1 (0.05%)
**Smoking Habits (N, %)**	
Past Smoker	410 (18.97%)
Current Smoker	334 (15.46%)
Never Smoker	1417 (65.57%)
**Body Mass Index (BMI) (N, %)**	
Underweight	50 (2.31%)
Normal weight	1229 (56.87%)
Overweight	678 (31.37%)
Obesity	191 (8.84%)
Unknown	13 (0.60%)
**Occupational Status (N, %)**	
Employed	1802 (83.39%)
Retired	196 (9.07%)
Student	92 (4.26%)
Unemployed	62 (2.87%)
Unknown	9 (0.42%)
**With whom he/she lives**	
Alone	357 (16.52%)
Not Alone	1718 (79.50%)
Unknown	86 (3.98%)
**Comorbidity**	
None	1613 (74.64%)
1	429 (19.85%)
2	102 (4.72%)
>2	17 (0.79%)
**Vaccinations**	
Anti-flu in last autumn	412 (19.07%)
Pneumococcal in the last 12 months	53 (2.45%)
Other vaccinations in the past 12 months	254 (11.75%)
**Symptoms**	
Diarrhoea	307 (14.2%)
Nausea/vomiting	152 (7.0%)
Poor appetite	236 (10.9%)
Smell dysfunction	279 (12.9%)
Taste dysfunction	280 (13.0%)
Dizziness	163 (7.5%)
Headache	641(29.7%)
Sleepiness during the day	369 (17.1%)
Confusion	95 (4.4%)
Involuntary tremors	80 (3.7%)
Tingling in the limbs/face	163 (7.5%)
At least one of previous symptoms	959 (44.4%)

**Table 2 ijerph-19-03058-t002:** Results from logistic regressions (univariate and multivariate) for the cohort of 959 interviewed participants reporting at least one symptom.

	Univariate Analysis		Multivariate Analysis ^1^	
Variable	OR (95% CI)	*p*-Value	OR (95% CI)	*p*-Value
Age				
<55	1.00 (Reference)			
>55	0.95 (0.67–1.35)	0.785		
**Gender**				
Female	1.00 (Reference)			
Male	0.96 (0.73–1.25)	0.749		
**Lives alone**				
No	1.00 (Reference)			
Yes	0.99 (0.69–1.43)	0.975		
**Smoking**				
Smoker	1.00 (Reference)			
Non-smoker	1.47 (0.99–2.18)	0.055		
Former smoker	1.26 (0.78–2.04)	0.338		
**Comorbidity/chronic condition**				
No	1.00 (Reference)			
Yes	0.90 (0.66–1.23)	0.506		
**Previous vaccinations**				
No	1.00 (Reference)			
Yes	1.16 (0.86–1.56)	0.322		
**Diarrhoea**				
No	1.00 (Reference)			
Yes	1.92 (1.44–2.54)	<0.001		
**Nausea/vomiting**				
No	1.00 (Reference)			
Yes	2.91 (2.04–4.14)	<0.001		
**Poor appetite**				
No	1.00 (Reference)		1.00 (Reference)	
Yes	6.02 (4.38–8.28)	<0.001	2.42 (1.63–3.62)	<0.001
**Smell dysfunction**				
No	1.00 (Reference)			
Yes	6.26 (4.62–8.49)	<0.001		
**Taste dysfunction**				
No	1.00 (Reference)		1.00 (Reference)	
Yes	7.17 (5.27–9.75)	<0.001	1.67 (1.20–2.34)	<0.001
**Dizziness**				
No	1.00 (Reference)			
Yes	1.76 (1.25–2.48)	0.013		
**Headache**				
No	1.00 (Reference)			
Yes	1.00 (Reference)	0.447		
**Sleepiness during day**				
No	1.00 (Reference)		1.00 (Reference)	
Yes	2.12 (1.61–2.79)	<0.001	4.15 (2.13–8.09)	0.002
**Confusion**				
No	1.00 (Reference)			
Yes	2.34 (1.53–3.59)	<0.001		
**Involuntary tremors**				
No	1.00 (Reference)			
Yes	3.18 (1.99–5.07)	<0.001		
**Tingling in the limbs/face**				
No	1.00 (Reference)			
Yes	1.25 (0.88–1.77)	0.217		
**Number of overall symptoms ^2^**				
1	1.00 (Reference)			
2	2.40 (1.48–3.88)	<0.001		
3	6.06 (3.75–9.79)	<0.001		
≥4	6.06 (3.75–9.79)	<0.001		

OR: odds ratio; CI: confidence interval; Odds ratio represents the probability of contacting a GP over probability of not contacting a GP. ^1^ In the table, only the statistically significant variables implemented in the multivariate model are reported. ^2^ Not evaluated in multivariate analyses due to multicollinearity issues.

## Data Availability

The data are available upon request to the authors.

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
