# Peer review of "Patients’ Engagement in Early Detection of COVID-19 Symptoms: An Observational Study in the Very Early Peak of the Pandemic in Italy in 2020"

_ijerph, 2022, doi:10.3390/ijerph19053058_

Round 1

Reviewer 1 Report

I congratulate the authors for this very interesting work. All study that provides some analysis of the impact of COVID-19 in the society is always welcome, in special when the health care systems, public health conditions and social impacts are considered.

I have some considerations about the manuscript that may help you to improve the scientific soundness. 

1) The application of univariate logistic regression models aid to understand the occurrence probability of a phenomena according to one or multiple independent variables. In particular, those models work better for quantitative variables instead of qualitative ones. 

According to the section "2.2 Materials" and the results in Table 2, I understood that most of the variables are qualitative ones. Since you are investigating the association between calling the GP and the characteristics/variables, I believe you should consider other association test (as 2 x 2 qui-square, for example). 

Otherwise, if you are sure (and/or have any other specific reasons) that the univariate logistic analysis is the best choice for this data analysis, I suggest you detail and clarify your reasons for the readers.

2) A suggestion of alternative statistical analysis you could consider is the univariate and multivariate correspondence analysis(<https://en.wikipedia.org/wiki/Correspondence_analysis>). I believe such analysis may provide better results and may also fit better to your data.

Author Response

1) The application of univariate logistic regression models aid to understand the occurrence probability of a phenomena according to one or multiple independent variables. In particular, those models work better for quantitative variables instead of qualitative ones. 

According to the section "2.2 Materials" and the results in Table 2, I understood that most of the variables are qualitative ones. Since you are investigating the association between calling the GP and the characteristics/variables, I believe you should consider other association test (as 2 x 2 qui-square, for example). 

Otherwise, if you are sure (and/or have any other specific reasons) that the univariate logistic analysis is the best choice for this data analysis, I suggest you detail and clarify your reasons for the readers.

Reply:

We thanks the reviewer for this observation: we are agree that among basic techniques for summarizing and analyzing categorical survey data,  contingency tables and Pearson’s Chi-squared tests are common used. Whilst these methods are a great way to start exploring the categorical data, to really investigate them fully, we decided to apply a more formal approach using generalized linear models. In particular, logistic regression models are designed for analyzing binary (e.g. yes/no) response variables and can accommodate continuous and/or categorical explanatory variables. Moreover, the chi-square test and univariate logistic regression gives the same results in term of statistical significance. The choice of the methodology is related to the aim of the study: the Chi-square test is really a descriptive test, akin to a correlation.  It’s not a modeling technique, so there is no dependent variable.  Since we want not only to describe the strength of a relationship but to model the determinants of and predict the likelihood of an outcome (the contact of GP), we retain that the logistic regression is appropriate. In the 2.3 Statistical analysis section, we added a sentence to clarify the use of this methodology.

2) A suggestion of alternative statistical analysis you could consider is the univariate and multivariate correspondence analysis(<https://en.wikipedia.org/wiki/Correspondence_analysis>). I believe such analysis may provide better results and may also fit better to your data.

Reply:

We thank the reviewer for this suggestion: the aim of the simple/multiple correspondence analysis is to identify a group of individuals with similar profile in their answers to the questions and to identify the associations between variable categories. As explained in the previous answer, our objective was to evaluate the impact of several symptoms on the contact of GP during the COVID 19 pandemic. For this study, then, we account more adequate the logistic regression models.

Reviewer 2 Report

Thank you for your submission. I hope you find my comments useful.

ABSTRACT: I recommend devoting more of your abstract towards giving actual results. As is, your abstract gives very little details about your sample and about what you actually found other than what you mentioned in lines 25-26.

INTRODUCTION: okay as-is, however, I found the background information a bit off topic and overly complex in regards to your actual research.

METHODS: good. I liked your strategy of first doing univariate analysis then taking relevant results and doing a multivariate analysis. However, I do think it would have been helpful to do an analysis of the entire group, rather than focusing just on the three sub-groups of blood donors, professionals, and volunteers.

RESULTS: good. My only recommendation would be to simplify the tables as much as possible, without removing important information. The tables you give are excellent in terms of detail, but hard to determine what is important and what is statistical noise.

DISCUSSION: my major question was why did only 1 in 3 contact the GP. How available were telephone consultations at the time of your study? Was it more a distrust of the GP or a fear of going out of quarantine? When you say in the methods on line 127/128 "whether they had called" their GP, were you referring to making an appointment face-to-face, just making a telephone call, or what exactly? If your survey didn't answer this, then what would a typical person in the region be referring to when they say the "called" their GP? Called the office and spoke with the staff? Called and spoke with a nurse? Called and actually spoke with the GP? Called and actually made a face-to-face appointment?

OVERALL: interesting survey. Good statistical analysis. Would like more details regarding what exactly is meant by "called their general practitioner" (line 127/128).

Thank you once again for your research and manuscript submission. 

Author Response

ABSTRACT: I recommend devoting more of your abstract towards giving actual results. As is, your abstract gives very little details about your sample and about what you actually found other than what you mentioned in lines 25-26.

Reply

We thank the reviewer for this observation:  in the revised version we integrated the abstract as regard as the results

INTRODUCTION: okay as-is, however, I found the background information a bit off topic and overly complex in regards to your actual research.

Reply

Thank you! We have removed some passages from the introduction that we felt were less pertinent to the topic addressed in our study.

METHODS: good. I liked your strategy of first doing univariate analysis then taking relevant results and doing a multivariate analysis. However, I do think it would have been helpful to do an analysis of the entire group, rather than focusing just on the three sub-groups of blood donors, professionals, and volunteers.

Reply

We thank the reviewer for this suggestion:  we performed the analysis for the entire group and the results are reported in the new Table 2 

RESULTS: good. My only recommendation would be to simplify the tables as much as possible, without removing important information. The tables you give are excellent in terms of detail, but hard to determine what is important and what is statistical noise.

Reply

We thank the reviewer for this suggestion: the previous Table 2, in the revised version, has been submitted as Supplementary material (Table S1), while in the manuscript the new Table 2 shows the results of the entire cohort, so that it is easier the reading.

DISCUSSION: my major question was why did only 1 in 3 contact the GP. How available were telephone consultations at the time of your study? Was it more a distrust of the GP or a fear of going out of quarantine? When you say in the methods on line 127/128 "whether they had called" their GP, were you referring to making an appointment face-to-face, just making a telephone call, or what exactly? If your survey didn't answer this, then what would a typical person in the region be referring to when they say the "called" their GP? Called the office and spoke with the staff? Called and spoke with a nurse? Called and actually spoke with the GP? Called and actually made a face-to-face appointment?

Reply

At the time of the study, in Italy, most GPs suspended face-to-face visits unless it was strictly necessary, and only on appointment: to the best of our knowledge, all GPs were available for telephone or videocall consultations. In Italy, calling the study of a GP usually refers to wither talk to a secretary to make an appointment for a face-to-face meeting, or (and in particular since the pandemic began) to actually talk to the physician itself for a phone/online consultation, as most of the time no other healthcare workers or staff are present in their offices.

In the survey, we actually asked them if they “got in touch with” their GP, regardless of whether it was via email, phone, videocall or else. We have amended lines 127/8 and other parts of our paper to make this more clear.

As for the reason why only 1 in 3 got in touch with their GP, it is up to speculation, as we have no direct data from our participants regarding this aspect.

OVERALL: interesting survey. Good statistical analysis. Would like more details regarding what exactly is meant by "called their general practitioner" (line 127/128).

Reply

Thank you for your feedback. We have replied to this concern above.

Reviewer 3 Report

Manuscript ID: ijerph-1606413

Title: Patients’ engagement in early detection of COVID-19 symptoms: an
observational study in the very early peak of the pandemic in Italy in 2020

In public health perspective, the healthcare management of COVID-19 is definitely the most interesting aspect. Authors worked on a very potential field which demands more attention from the other groups of scientists all over the world. Such reports are very helpful to decipher the preventive measures against any pandemic or epidemic. The manuscript, in this sense, is of great interest. The results of this very work are important; and certainly suitable for publication. However, authors should extensively edit the manuscript in terms of English language correction to improve the readability of the paper. Here are some specific comments:

Lines 16-17: “COVID-19 exerted a strong impact on the Italian healthcare systems, which in turn resulted in a disruption of the citizens’ trust towards healthcare providers and institutions.”

-----“ disruption of the citizens’ trust” ---- I don’t understand what the authors meant actually. A pandemic is always dreadful. Authors better replace these group of words with something else like “resulted in a confusion towards healthcare providers….”

Line 20-23: Can authors clary the aim of this study more clearly? The sentence may appear to be too long to understand for the wide range of readers.

Lines 33-34: The first sentence of the Introduction is carrying a negative meaning for the healthcare system. Authors may delete this sentence. And they should include the effort by the healthcare providers. Challenges should come up in the next sentence. Therefore, authors need to revise the starting of the Introduction.

Lines 43-44: “……less urgent and worth of attention symptoms and signs not specifically connected to COVID-19”-------- Not clear to me.

Lines 54-58: “This decreased sense of agency and self-effectiveness in successfully managing one’s own health and life-style is a preoccupying element which may undermine the level of vigilance of patients on their symptoms and their willingness to promptly communicate to the reference healthcare professionals the eventuality of new symptoms or signs” ---- Authors should rephrase this sentence with a complete meaning.

Lines 62-63: “An additional cause of the diffused sense of impotence and fatalism in the society 62 during the pandemic period is the critical level of trust in science.”

Where did the author find the word “trust”? What does it mean? Ethics or consent? Reliability? Authors must clarify and fix this sentence with an appropriate reference.

Lines 65 and 70 also need to be corrected in terms of the word “trust”. The entire manuscript should be checked for this superficial term.

Lines 72-75: Any reference for this sentence? Or this is the conclusive imagination from authors? Please re-write this sentence with a clear meaning.

Line 80: Please include a relevant reference.

Line 85: “Paradoxically, thus, although ……..” please don’t start a sentence this way. I think the authors should edit the manuscript for language correction by a native English speaker.

Lines 93-96: Please merge these sentence into the previous paragraph.

Lines 114-126: Omit the bullets; and write in a sentence.

Lines 130-132: Be more specific in the statistical analysis method with relevant reference.

Lines 163-166: English grammar should be corrected.

Lines 168-175: Why are the three sentences in three separate paragraphs? Authors should fix such errors throughout the manuscript.

Lines 187-188: “Finally, as for the volunteers, the univariate analyses showed that diarrhoea, nausea/vomit, poor appetite, smell dysfunctions, taste dysfunctions, and involuntary tremors”----- Not “vomit”, it should be “vomiting”. Not smell dysfunctions; rather it should be dysfunction; and so for the taste dysfunction.

Lines 195-196: I didn’t understand the meaning of this sentence. Authors should rephrase.

Lines 224-228: Too long sentence.

Lines 256-257: No need to create a paragraph for a single sentence.

Lines 261-262: Avoid the repetition what has been stated earlier.

Lines 262-263: Why do the authors consider the data to be surprising? That’s normal. That’s expected. Please rephrase that portion.

Lines 270-274: Too long sentence. Thais is the common problem of this manuscript. This decreases the readability interest. Authors should use simple sentences.  

Line 280: The word “for” is not needed in “……………thus claim for a better consideration of the importance of engaging patients in their………”

Round 2

Reviewer 1 Report

Dear authors, thank you for your responses.